# Low-Ammonium Environment Increases the Nutrient Exchange between Diatom–Diazotroph Association Cells and Facilitates Photosynthesis and N_2_ Fixation—a Mechanistic Modeling Analysis

**DOI:** 10.3390/cells11182911

**Published:** 2022-09-17

**Authors:** Meng Gao, Gabrielle Armin, Keisuke Inomura

**Affiliations:** Graduate School of Oceanography, University of Rhode Island, Narragansett, RI 02882, USA

**Keywords:** DDA, ammonium, nutrient exchange, nitrogen fixation, photosynthesis, diatom, diazotroph, carbon, nitrogen, cell flux model

## Abstract

Diatom–diazotroph associations (DDAs) are one of the most important symbiotic dinitrogen (N_2_) fixing groups in the oligotrophic ocean. Despite their capability to fix N_2_, ammonium (NH_4_^+^) remains a key nitrogen (N) source for DDAs, and the effect of NH_4_^+^ on their metabolism remains elusive. Here, we developed a coarse-grained, cellular model of the DDA with NH_4_^+^ uptake and quantified how the level of extracellular NH_4_^+^ influences metabolism and nutrient exchange within the symbiosis. The model shows that, under a fixed growth rate, an increased NH_4_^+^ concentration may lower the required level of N_2_ fixation and photosynthesis, and decrease carbon (C) and N exchange. A low-NH_4_^+^ environment leads to more C and N in nutrient exchange and more fixed N_2_ to support a higher growth rate. With higher growth rates, nutrient exchange and metabolism increased. Our study shows a strong effect of NH_4_^+^ on metabolic processes within DDAs, and thus highlights the importance of in situ measurement of NH_4_^+^ concentrations.

## 1. Introduction

Diatom–diazotroph associations (DDA) are one of the major symbiotic N_2_ fixing groups in the low-nutrient ocean [1,2,3,4,5], which are composed of one diatom host (common genera: *Hemiaulus*, *Rhizosolenia*, and *Chaetoceros*) and symbiotic cyanobacterial diazotrophs (hereafter “diazotrophs”; common genera: *Richelia* and *Calothrix*) [6,7,8,9,10,11,12]. The symbiotic diazotrophs can form trichomes including heterocysts (cells specialized in N_2_ fixation that cannot fix C, one heterocyst in one trichome) and vegetative cells (photosynthesis cells) [13]. Previous modeling research suggested that a significant amount of N is transferred from the diazotroph to the host diatom [4]. This high level of N_2_ fixation by the diazotrophs is possibly enabled by their metabolic pathways [14]. In addition, a recent modeling study showed that a high rate of N_2_ fixation was enabled by a significant amount of C transfer from the host diatom [6], and similar results were reported in recent in situ measurements [11].

These nutrient transfers are also implied in other symbiotic associations, for example, unicellular cyanobacterium (UCYN-A) and haptophyte [15], Rhizobium and legumes [16,17], and *Azolla–Anabaena* symbiosis [18]. Studies on these other symbioses on genetic [19] and metabolic levels [18] also suggested that C and N fluxes in nutrient transfer are closely related to metabolic processes, such as photosynthesis and N_2_ fixation, which respectively contribute to the C and N sources. However, these metabolic processes are not the only sources of C and N in the natural ocean; environmental nutrients are also important.

Nutrient availability is crucial in DDA research. Nitrogen, phosphorus, iron, and silicon are all reported to control DDA blooms in the natural ocean, such as in the North Pacific [20,21,22], tropical Atlantic [23], and the Red Sea [24]. In addition to these observations, some modeling studies suggest that DDA blooms and distributions are related to nutrient limitation [25,26,27]. For example, DDA blooms in the Amazon River plume can be driven by the N-poor and Si-rich water [25], and the global DDA distribution can be controlled by dissolved iron and phosphate concentrations [26,27]. Ammonium (NH_4_^+^), one of the key phytoplankton N sources [28], has been widely observed to influence diatom growth and blooms [29,30,31]. A culture study reported that ammonium concentration affected diatom nutrient uptake of other nutrients, such as sulfide and carbon [29]. Additionally, an observation in a shallow estuary revealed that a low NH_4_^+^ concentration characterized the initial period of a diatom bloom [31]. As an extracellular source of N, environmental NH_4_^+^ can influence growth and metabolism in various non-symbiotic diazotrophs. For a unicellular diazotroph, *Crocosphaera watsonii* [32], increasing environmentalNH_4_^+^ concentration can decrease its growth rate, and for filamentous diazotrophs, such as *Trichodesmium*, the use of environmental NH_4_^+^ can increase the growth rate [33].

How does NH_4_^+^ concentration influence symbiotic diazotrophs, such as DDAs? Despite its potential significance in their metabolism and their outcome in the environment, there are limited studies on this topic. A recent study reported that added NH_4_^+^ is indeed consumed by DDAs, but they exhibited similar growth to those in diazotrophic conditions [34]. According to observations in the North Pacific Subtropical Gyre [35] and Mediterranean Sea [36], lower N can lead to more symbiotic diatoms. Symbiosis compensates for the lack of nutrients by offering nutrient exchange, which leads to a higher growth rate [6]. In our study, by quantifying the effect of NH_4_^+^ availability on nutrient exchange and metabolism, we offer a metabolic-level implication for why DDAs are abundant in the oligotrophic ocean.

To predict how environmental NH_4_^+^ influences the inner element (C and N) flux and metabolic reaction rate, we built a mechanistic model of DDAs (Figure 1) based on a previous *Hemiaulus* (diatoms)–*Richelia* (diazotrophs) model [6] and included NH_4_^+^ as an extracellular nutrient source. The model assumes that the system is in a steady state, with element supply equal to consumption. The C supply includes photosynthesis in diatoms and vegetative cells, and the N supply includes N_2_ fixation in heterocysts and NH_4_^+^ uptake. C consumption includes biosynthesis and C usage in N_2_ fixation, and N consumption includes biosynthesis. We mainly focused on the C and N transfer and metabolism rates. We also considered other factors, such as the number of trichomes, growth rate, and light intensity, to make the model closer to reality. Our model provides quantitative answers to the following questions: (1) How does the NH_4_^+^ concentration influence nutrient exchange between the host diatom and trichomes? (2) How does the NH_4_^+^ concentration influence cell metabolism, such as photosynthesis and N_2_ fixation? (3) How do the NH_4_^+^ concentration and other factors influence nutrient exchange and metabolism together? (4) How does the NH_4_^+^ concentration alter the elemental fate?

## 2. Results and Discussion

### 2.1. NH_4_^+^ Uptake Influences Nutrient Exchange and Metabolism

NH_4_^+^ is an essential environmental N species in the water. We considered it as another N supply separate from N_2_ fixation. We resolved the relationship between the NH_4_^+^ concentration and the uptake rate based on previous data from diatom studies (Equation (4), [28,37]). Here, we fixed the growth rate at 0.51 d^−1^, the average value observed [38] and used in a previous model [6]. The model predicted the steady-state metabolism, where the supply of fixed C and N is used without any waste to maximize growth. Similar assumptions were made in previous studies [39,40,41,42,43,44]. Additionally, a recent culture study of DDAs shows that most of the provided NH_4_^+^ was consumed [34], supporting the assumption.

The model showed that more C can be transferred in an environment with a lower NH_4_^+^ concentration (Figure 2a). The amount of transferred C increased from 11.1% to 22.7% when the NH_4_^+^ concentration decreased from 0.04 mmol m^−3^ to 0 mmol m^−3^ (Figure 2a). The net C transfer is consistent from the diatom to trichome. On the other hand, as the NH_4_^+^ concentration increases, less N is transferred from the diatoms to the trichomes (Figure 2b). When the NH_4_^+^ concentration is high enough to support, or even exceeds, the consumption requirement by the diatom, the transfer direction changes, bringing excess NH_4_^+^ from the diatoms to the trichomes. When NH_4_^+^ uptake is equal to the consumption of N by the diatom, there is no transfer (Figure 2b, dash line). We name this value as the no transfer NH_4_^+^ concentration, which we calculated (Appendix A) to be 0.034 mmol m^−3^ for the assumed growth rate.

The model shows that the NH_4_^+^ concentration also influenced metabolic processes, including photosynthesis and N_2_ fixation. Our results show that, with a fixed growth rate, photosynthesis and C transfer are higher with lower NH_4_^+^ concentrations. (Figure 2a,c). The model analysis suggests that these occur because, with less NH_4_^+^, a higher N_2_ fixation rate is necessary to support the N supply (Figure 2d), which requires more fixed C to provide both energy and electrons. This allows more N to be transferred to the diatoms to compensate for the lack of N. On the other hand, when NH_4_^+^ uptake can supply all the N required for consumption in the entire DDA system, there is no N_2_ fixation.

Moreover, our results show that DDAs in low-nutrient areas need more nutrient (C and N) transfer to maintain a fixed growth rate, which corresponds to studies reporting the nutrient transfer and higher N_2_ fixation rate in DDAs [4,6,38]. This nutrient transfer, facilitated by N_2_ fixation, may be the reason why symbiosis occurs in low-nutrient habitats. Compared with non-symbiotic diazotrophs (nutrients are all from uptake and themselves), DDAs with nutrient transfer can maintain a faster growth rate in the oligotrophic ocean and even form seasonal blooms in some ocean areas [2,4,11,38].

### 2.2. Effect of Growth Rate

In the above simulations, we fixed the growth rate at an average value. However, in nature, growth rates vary. The growth rate determines how much C and N DDAs require for biosynthesis. Thus, it influences nutrient consumption, altering nutrient exchange and nutrient-related metabolism. We set the value of the growth rate within a reasonable range (0.3 d^−1^–0.8 d^−1^) [6,45,46,47], kept the same range for NH_4_^+^, and ran the simulation (Figure 3). Our results show that, with a higher growth rate and lower NH_4_^+^ concentration, more C and N are transferred (Figure 3a,b). To support more consumption under higher-growth-rate conditions, photosynthesis in diatoms (Figure 3c) and N_2_ fixation also increased (Figure 3d).

This result is consistent with the result from the DDA-CFM model [6], which showed significant C flux from the diatom enhanced by both the growth and N_2_ fixation rates. According to the results, we can also suggest that, at a certain nutrient level, a higher growth rate frequently corresponds to stronger nutrient exchange (Figure 3a,b), which can be achieved by symbiosis. This metabolic connection can explain why non-symbiotic diazotrophs grow at only approximately 0.3 d^−1^ [6,46,47,48], whereas symbiotic diazotrophs can grow at a rate as high as 0.87 d^−1^ under diazotrophic conditions [6,45].

The model results show that the growth rate and NH_4_^+^ concentration can also have some interactions. Specifically, the growth rate can weaken the influence of the NH_4_^+^ concentration: C and N transfers decrease slower with the NH_4_^+^ gradient under higher growth rates (Figure 3a,b); additionally, the positive effect of the growth rate on the metabolism rate (Figure 3c,d) is opposite to the negative effect of the NH_4_^+^ concentration. The opposite effect of the growth rate and NH_4_^+^ concentration results from the different roles they play. NH_4_^+^ is nutrient supply, while growth is nutrient consumption; thus, these variables appear on opposite sides of the balance Equation (3).

### 2.3. Element Fate and Flux in Different NH_4_^+^ Concentrations and Growth Rates

Here, we compared the C and N fates under different NH_4_^+^ concentrations (0.01 mmol m^−3^, 0.036 mmol m^−3^) and growth rates (0.4 d^−1^, 0.8 d^−1^) to understand how varying environmental and cellular conditions impact symbiosis. In a lower-NH_4_^+^ environment (Figure 4a,b), the model shows that higher portions of C and N are transferred from diatom to trichomes (Figure 4a (19%), b (21%) compared with Figure 4c (9%), d (16%), Figure 5a). Additionally, under lower-NH_4_^+^ environments, C transfer provides more trichomes with C to use compared with C generated by photosynthesis within the trichomes (Figure 4a,b). In a lower-NH_4_^+^ environment, N_2_ fixation in heterocysts is the main N source (Figure 4a,b, 71% and 85%). To achieve a higher growth rate (0.8 compared with 0.4), the model suggests that more N_2_ fixation (14% higher compared with lower-growth-rate conditions, Figure 4a,b and Figure 5b) is needed. C and N transfer are also higher (C: 2% higher, N: 14% higher, Figure 4a,b and Figure 5a) to support the higher growth rate. These results are consistent with the above simulation, which indicates that a lower NH_4_^+^ environment has a higher nutrient exchange and more N_2_ fixation. They also suggest that, in oligotrophic areas, nutrient exchange between cells and N_2_ fixation in trichomes can make the DDA an efficient system to support a higher growth rate.

In a higher-NH_4_^+^ environment (Figure 4c,d), the model shows that if NH_4_^+^ is high enough to support all of the needs (e.g., when the growth rate is 0.4 d^−1^ and NH_4_^+^ concentration is 0.036 mmol m^−3^, Figure 4c and Figure 5b), N_2_ fixation is unnecessary. However, if it needs to achieve a higher growth rate (Figure 4d), the model indicates that N_2_ fixation and element transfer still need to increase. Compared with a lower-NH_4_^+^ environment (Figure 4c,d compared with Figure 4a,b and Figure 5), the increasing portion of N_2_ fixation and element transfer is larger with the same increasing growth rate (0.4 d^−1^ to 0.8 d^−1^) under the higher-NH_4_^+^ environment. In the low-NH_4_^+^ environment, the model shows that N_2_ fixation increases by 14% (Figure 4a,b and Figure 5b), while in the high-NH_4_^+^ environment, it increases by 50% (Figure 4c,d and Figure 5b). For C transfer, it increases by 2% in the low NH_4_^+^ environment (Figure 4a,b and Figure 5a), while in the high-NH_4_^+^ environment, it increases by 7% (Figure 4c,d and Figure 5a). This result is similar to that in Figure 3, which shows that nutrient exchanges and N_2_ fixation increase faster in a higher-NH_4_^+^ environment (Figure 3a,b). These results suggest that, in a higher-NH_4_^+^ environment, DDAs need more nutrient exchange and N_2_ fixation to reach a higher growth rate. Thus, we may assume that the advantage of symbiosis is less obvious in nutrient-rich oceans than in oligotrophic oceans, because nutrient exchange within symbiosis can be more useful when nutrients are scarce. Furthermore, the results are consistent with previous research that, in some nutrient-rich areas, non-symbiotic diatoms are more abundant [5].

### 2.4. Effect of Other Factors: Trichome Number and Light Intensity

In natural conditions, other factors, such as DDA characteristics (number of trichomes) and environmental factors (e.g., light intensity), also vary [4]. Here, we considered the number of trichomes (1 to 5, [4,6,49,50]) and light intensity in the model.

#### 2.4.1. Trichome Number

Here, we simulated the effect of different trichome numbers (1 to 5) on element transfer and metabolism (Appendix A). Because an increasing number of trichomes increases their C and N consumption, more C is transferred from diatoms to trichomes (Appendix A) to support the need for trichomes, and less N is transferred to the diatom (Appendix A), since trichomes consume more. The model suggests that, to support this high consumption, the photosynthesis rate (for the whole DDA, diatom, and trichomes) and N_2_ fixation rate also increase with an increasing number of trichomes (Appendix A).

As for the changing trend with NH_4_^+^ concentration, with more trichomes, nutrient exchange changed more slowly (Appendix A). From the perspective of nutrient transfer, diatoms with fewer trichomes are more sensitive to the change in the environmental N concentration, since more trichomes have higher rates of N_2_ fixation (Appendix A) to compensate for the lack of N in the DDA system. The weakened effect of the trichome number on NH_4_^+^ is similar to the effect of growth rate (Figure 3), since they both increase the consumption of elements.

#### 2.4.2. Light Intensity

As one of the most critical limiting factors in the ocean, light is the energy source of photosynthesis [51,52]. Since previous studies reported that light influences diatoms [53,54] and cyanobacteria [55,56,57], here, we also tested the effect of light intensity on the symbiotic metabolisms and nutrient exchanges in DDAs. The model result shows that, when light intensity increases, the photosynthesis rate and C transfer also increase (Appendix A). However, the N pathways are not influenced by light intensity (Appendix A). In a previous study, researchers also reported no significant relationship between the vertical distribution of cyanobionts (cyanobacteria symbionts) and light levels [5], which can be due to the small effect on N pathways and exchanges. However, in some other studies, opposite to our results, N_2_ fixation can be connected to the light intensity, since it can be fueled by C fixation relating to the light [11,58].

### 2.5. Comparison to Previous Studies and Implications for Future Work

In the previous DDA model [6], N_2_ fixation was considered as the only N supply. However, in the real world, various species of nutrients can also influence the association between DDA cells and nutrient-related metabolism. Our study included one of the extracellular nutrient sources and discussed its influence. In the future, we can include more nutrient sources in the DDA model. For example, nitrate is another common nutrient species in the ocean biochemical cycle and is another N source [59,60,61,62]. Including more nutrient species can make the model results closer to the natural condition and easier to be compared with real data. This model also predicts that lower NH_4_^+^ increases element transfer and enhances metabolic processes with a fixed growth rate. Based on this, we can offer a possible explanation for the reason that cell connections with nutrient exchange are common in low-nutrient habitats. These model-based predictions and hypotheses can be further tested with additional in situ measurements and observational data. Our study can also be complemented by omics analysis [63,64] to further explore how metabolism and nutrient exchanges may change under various NH_4_^+^ concentrations.

## 3. Conclusions

By including NH_4_^+^ as another N source in the DDA model, according to our simulation results, an increased NH_4_^+^ concentration may lower the required level of N_2_ fixation and photosynthesis and decrease C and N exchange under a fixed growth rate. With a higher growth rate, nutrient exchange and metabolism increase. A low-NH_4_^+^ environment uses more C and N in nutrient exchange and more N_2_ fixation to support a higher growth rate, which means that a stronger connection (higher nutrient exchange) between the cells in DDAs is necessary. With an increased number of trichomes, C transfer increases while N transfer decreases, and metabolism increases. With increased light intensity, C transfer and photosynthesis increase while N transfer and N_2_ fixation do not change. Increasing DDA consumption, such as the growth rate and trichome number, can weaken the effect of NH_4_^+^ because more N_2_ can be fixed by trichomes. Our study shows a strong effect of NH_4_^+^ on nutrient exchange and metabolic processes within DDAs. These results can better our understanding of the DDA nutrient flux in oligotrophic oceans and highlight the importance of environmental NH_4_^+^.

## 4. Materials and Methods

The DDA model is based on the following equations representing the balance of element supply and consumption. To obtain these equations, we assumed that each C and N pool is in a steady state. We considered processes including photosynthesis, N_2_ fixation, biosynthesis, and NH_4_^+^ uptake in our model. The following equations are based on equations from a previous study [6]. We include additional information regarding the derivation of equations and parameter definitions Appendix A.

### 4.1. C Balance

Under the steady state, we obtained the balance of C metabolism (Equation (1)) (See Appendix A for its derivation). The equation describes the balance between C supply and consumption. Here, the C supply includes two sources, C fixation (photosynthesis) in the diatom and in vegetative cells ($F_{pho}^{D}$ and $F_{pho}^{V}$, unit pmol C d^−1^ cell^−1^, mean the daily rate of per-DDA photosynthesis by diatoms and vegetative cells, respectively). C fixation in vegetative cells has been demonstrated in a previous study [11]. Consumption includes growth ($\mu \;\left(Q_{C}^{V}+Q_{C}^{H}+Q_{C}^{D}\right)$, this term means the C usage in growth, unit pmol C d^−1^ cell^−1^), respiration ($\mu \;\left(Q_{C}^{V}+Q_{C}^{H}+Q_{C}^{D}\right)\;E$, this term means the C usage in respiration, unit pmol C d^−1^ cell^−1^), and N_2_ fixation ($F_{C}^{N_{2}fix}$, unit pmol C d^−1^ cell^−1^), where *µ* is the growth rate (d^−1^), and the host diatom, vegetive cells, and heterocysts grow at the same rate, according to previous experimental studies on genomic analysis [65,66]. $Q_{C}^{V}$, $Q_{C}^{H}$, and $Q_{C}^{D}$ are the cellular C quotas of vegetative cells, heterocysts, and diatom per DDA, which were calculated from experimental data of the cell volumes [4] and the C to volume relationship [67], and *E* is the ratio of respiration to biosynthesis [6,68]. Equation (2) is used to calculate the C usage in N_2_ fixation ($F_{C}^{N_{2}fix}$, unit pmol C d^−1^ cell^−1^), which equals the N usage in N_2_ fixation ($F_{N}^{N_{2}fix}$, unit pmol N d^−1^ cell^−1^) multiplied by the C to N cost ratio in N_2_ fixation ($F_{C:N}^{N_{2}fix}$, unit pmol C pmol N^−1^).
(1)$$F_{pho}^{D}+F_{pho}^{V}=\mu \left(Q_{C}^{V}+Q_{C}^{H}+Q_{C}^{D}\right)\left(1+E\right)+F_{C}^{N_{2}fix}$$
(2)$$F_{C}^{N_{2}fix}=F_{N}^{N_{2}fix}\times F_{C:N}^{N_{2}fix}$$

### 4.2. N Balance

Similarly, under the steady state, we obtained Equation (3), which describes the balance between N supply and consumption (see the Appendix A for the derivation). Here, the N supply includes two sources, N_2_ fixation by heterocysts ($F_{N}^{N_{2}fix}$, unit pmol N d^−1^ cell^−1^), and NH_4_^+^ uptake by diatoms ($V_{NH_{4}^{+}}$, unit pmol N d^−1^ cell^−1^). All this N is used in growth ($\mu \left(Q_{N}^{V}+Q_{N}^{H}+Q_{N}^{D}\right)$), where *µ* is the growth rate (d^−1^), and $Q_{N}^{V}$, $Q_{N}^{H}$, and $Q_{N}^{D}$ are the cellular N quotas (unit mol N cell^−1^) of the vegetative cells, heterocysts, and diatoms per DDA. We calculated the NH_4_^+^ uptake by using Equation (4) to make the NH_4_^+^ uptake a function of the NH_4_^+^ concentration ($[NH_{4}^{+}],$ unit mmol m^−3^). The NH_4_^+^ uptake rate ($V_{NH_{4}^{+}}$, unit pmol N d^−1^ cell^−1^) can be faster with higher environmental NH_4_^+^, but it will come to saturation when reaching a maximum ($V_{NH_{4}^{+}}^{max}$, unit pmol N d^−1^ cell^−1^), so we used a function resembling Monod Kinetics. Here, $K_{m}$ is the half-saturation concentration (unit mmol m^−3^).
(3)$$F_{N}^{N_{2}fix}+V_{NH_{4}^{+}}=\mu \left(Q_{N}^{V}+Q_{N}^{H}+Q_{N}^{D}\right)$$
(4)$$V_{NH_{4}^{+}}=V_{NH_{4}^{+}}^{max}\frac{\left[NH_{4}^{+}\right]}{[NH_{4}^{+}]+K_{m}}$$

### 4.3. Values and Calculations

In Equation (1), we calculated the $Q_{C}^{V}$, $Q_{C}^{H}$, and $Q_{C}^{D}$ values following a method reported in a previous paper [6]. We used typical cell volumes (3493.5 µm^3^ for a diatom, 18.8 µm^3^ for a vegetative cell, and 61.0 µm^3^ for a heterocyst) [4], reported relationships between cell volume and C quotas [67], and typical cell ratios (diatom to trichomes: 1:2 and vegetative cells to heterocysts: 4:1, from observations on *Hemiaulus* and *Richelia* relationships) [4,14,50,69] to calculate them. Then, based on the Redfield ratio [70] (an empirical value that was used in the previous DDA study [4], and C:N is considered as 6.6:1), we converted the C quotas to N quotas ($Q_{N}^{V}$, $Q_{N}^{H}$, and $Q_{N}^{D}$). The scale of the growth rate (0.3–0.8 d^−1^, consistent with the previous experimental data [45,46,47,48,69,71,72]), and the value of E (0.38) were also obtained from the previous paper [6]. To test the sensitivity of the E value, we conducted a sensitivity test by doubling the E value in the vegetative cells (Appendix A). The results are similar to those in the main text with the default E values, thus suggesting that our overall conclusion is robust.

$F_{C}^{N_{2}fix}$ in Equation (1) was calculated based on Equations (2)–(4). We obtained $V_{NH_{4}^{+}}^{max}$ (1.16 pmol N C^−1^ d^−1^, multiplied by cellular C quotas to convert to pmol N cell^−1^ d^−1^) and $K_{m}$ (0.483 mmol m^−3^) from a previous NH_4_^+^ uptake modeling paper [28]; these values are within a reasonable range [73]. Then, $F_{N}^{N_{2}fix}$ can be solved by Equation (3) and $F_{C}^{N_{2}fix}$ can be solved by Equation (2).

Since we already have all of the values on the right side of the Equation (1), the value of $F_{pho}^{D}+F_{pho}^{V}$ can be solved. Then, we assumed that the rates of photosynthesis are proportional to the cellular N quotas and obtained the $F_{pho}^{D}$ and $F_{pho}^{V}$ values. When we considered light intensity as another influencing factor, we did not use this method to calculate photosynthesis. We used Appendix A to consider photosynthesis as a function of light intensity. We also conducted a sensitivity test by lowering 50% of the vegetative cell’s maximum photosynthesis rate (Appendix A). The result was similar to the results mentioned in Section 2.4.2 and Appendix A, suggesting that the conclusion regarding light intensity is robust.

## Figures and Tables

**Figure 1 cells-11-02911-f001:**
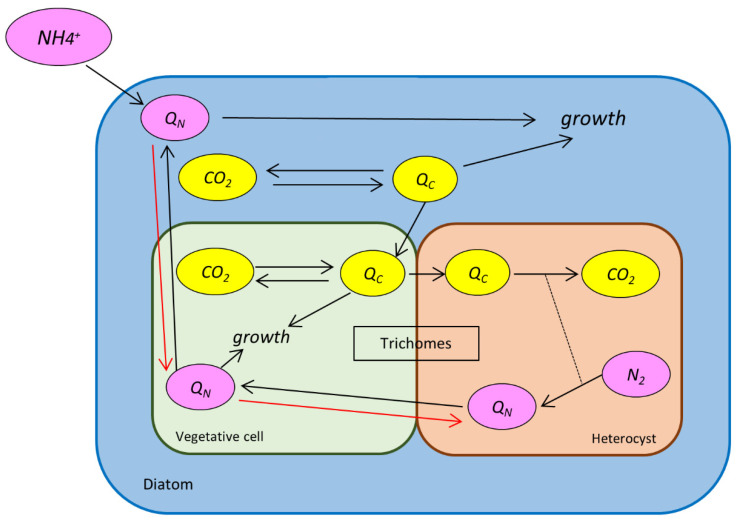
Schematic of the cell flux model of the diatom–diazotroph association (DDA). Blue frame and space: the host diatom. Green space and green frame: vegetative cell in trichomes. Brown frame and light-brown space: heterocyst in trichomes. Yellow ovals: C pools. Pink ovals: N pools. Black arrows: direction of the element flux when the NH_4_^+^ supply is not enough for N consumption in the host diatom cell. Red arrows: direction change when the NH_4_^+^ supply is higher than the diatom consumption. Dashed line: coupling between the processes.

**Figure 2 cells-11-02911-f002:**
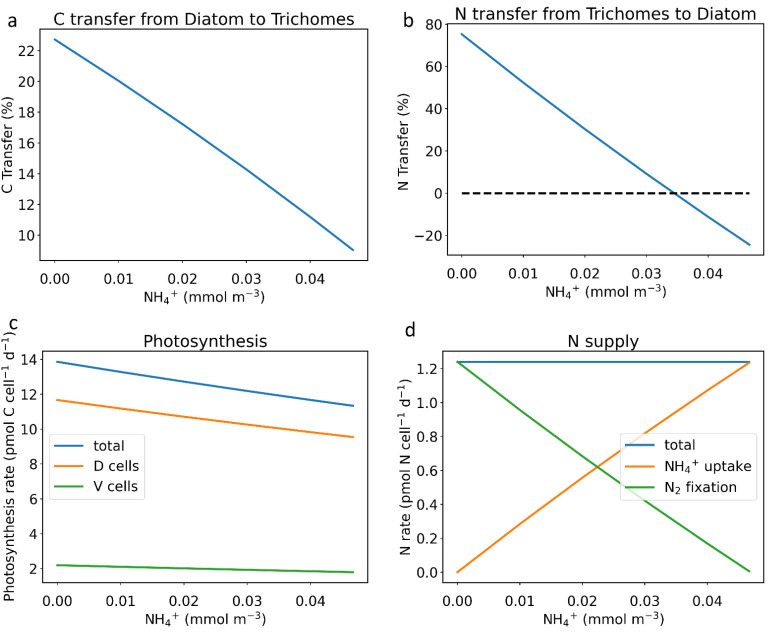
Simulated influences of NH_4_^+^ concentration on element supply and transfer. (**a**) Effect of NH_4_^+^ on C transfer. (**b**) Effect of NH_4_^+^ on N transfer; the dashed line is the no transfer NH_4_^+^ concentration. (**c**) Effect of NH_4_^+^ on photosynthesis; the blue line is the photosynthesis change in DDA, orange line is the photosynthesis change in the host diatom, and green line is the photosynthesis change in the vegetative cell. (**d**) Effect of NH_4_^+^ on N supply the blue line is the total N supply change, orange line is the NH_4_^+^ uptake change, and green line is the N_2_ fixation change. For (**a**,**b**), the unit percentage means how much C and N transfer account for the total C and N consumption (or supply).

**Figure 3 cells-11-02911-f003:**
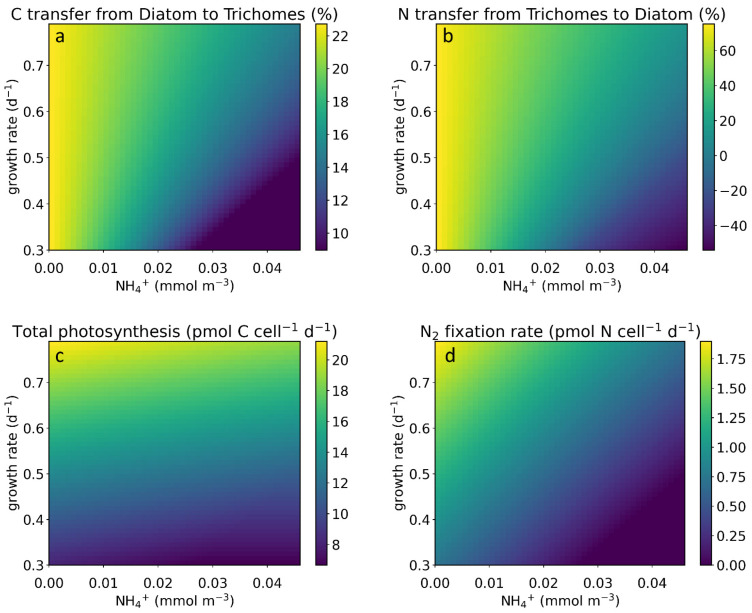
Simulated influence of the NH_4_^+^ concentration and growth rate on element transfer and supply. (**a**) Effect of NH_4_^+^ and growth rate on C transfer. (**b**) Effect of NH_4_^+^ and growth rate on N transfer. (**c**) Effect of NH_4_^+^ and growth rate on photosynthesis. (**d**) Effect of NH_4_^+^ and growth rate on N_2_ fixation. For (**a**,**b**), the unit percentage means how much C and N transfer account for the total C and N consumption (or supply).

**Figure 4 cells-11-02911-f004:**
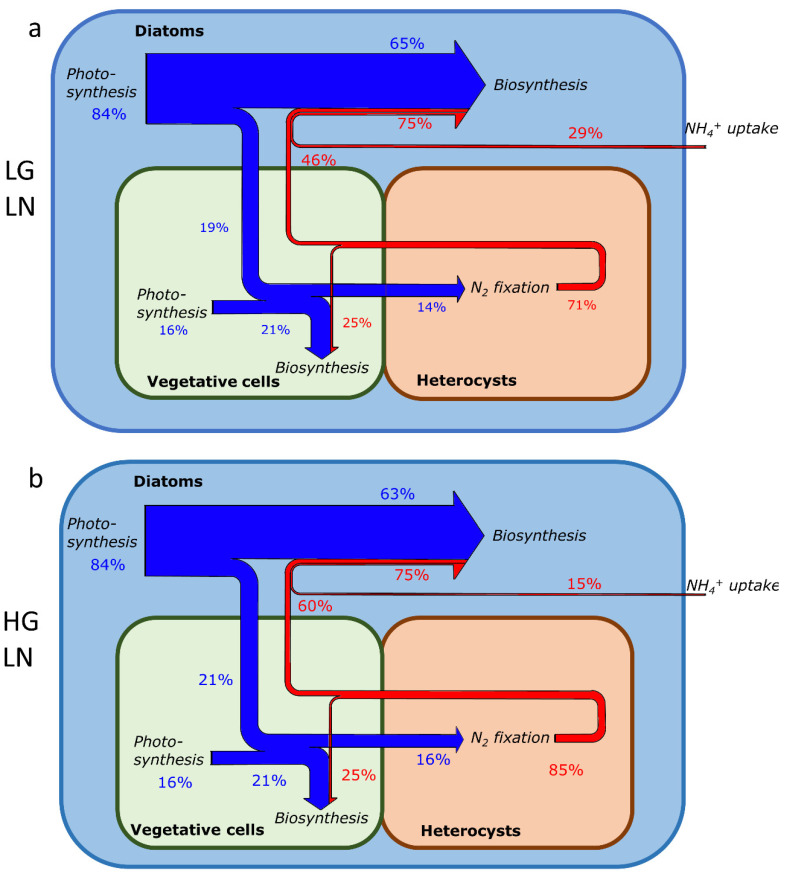
Elemental flux change with growth rate and NH_4_^+^ concentration. (**a**) Elemental flux under low-growth-rate and low-NH_4_^+^-concentration (LGLN) conditions. (**b**) Elemental flux under high-growth-rate and low-NH_4_^+^-concentration (HGLN) conditions. (**c**) Elemental flux under low-growth-rate and high-NH_4_^+^-concentration (LGHN) conditions. (**d**) Elemental flux under high-growth-rate and high-NH_4_^+^-concentration (HGHN) conditions. The 100% for this percentage value is the total C (blue arrows) and N (red arrows) supply (or consumption).

**Figure 5 cells-11-02911-f005:**
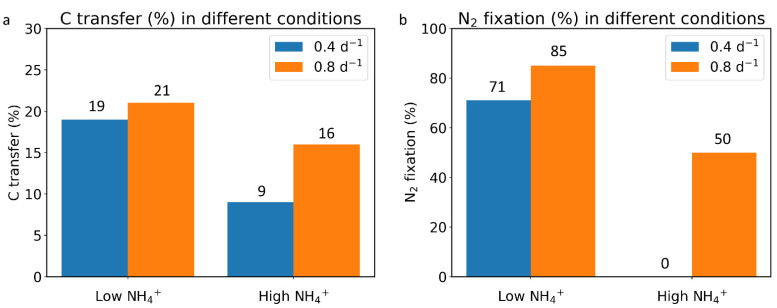
C transfer and N_2_ fixation under different conditions: low NH_4_^+^, high NH_4_^+^, low growth rate (blue bars), and high growth rate (orange bars). (**a**) C transfer. (**b**) N_2_ fixation.

## Data Availability

Our model is freely available from Zenodo at https://doi.org/10.5281/zenodo.6868690 (accessed on 1 July 2022).

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
