# Peer review of "Low-Ammonium Environment Increases the Nutrient Exchange between Diatom–Diazotroph Association Cells and Facilitates Photosynthesis and N_2_ Fixation—a Mechanistic Modeling Analysis"

_cells, 2022, doi:10.3390/cells11182911_

Round 1

Reviewer 1 Report

In the manuscript cells-1878456, Gao et al investigated interactions between diatoms and cyanobacteria under the relationship that the former is the host and latter is the symbiont. The authors examined effects of ammonium ion concentration on the flows of C and N between the diatoms and the cyanobacteria, and concluded that under a fixed growth rate, higher ammonium ion concentration lowers the required level of nitrogen fixation and photosynthesis and decreases C and N exchanges. The subject is surely interesting, therefore it seems suitable for the audience of the journal “Cells”, but not adequate for the special issue, because the present manuscript focuses only on the interaction between microbes, not between plant and microorganisms.

Introduction is not informative. How are nutrients exchanged between hosts and symbionts in symbiotic associations other than diatom-diazotroph associations? Many references are listed, but what was found by each reference is not clear. What did the previous studies reveal? 

Through the calculations, the authors assumed that the growth rate and the ratio of respiration to biosynthesis are constants for diatom, vegetative cells, and heterocyst. These assumptions seem to be too simplified to describe the balances between carbon supply and consumption and between nitrogen supply and consumption. Do the authors have any experimental evidences to support the assumptions?

Lines 106-108. Please explain here how the vegetative cells were confirmed to be photosynthetically active in diatom-diazotroph symbiosis.

Line 167. What do the authors exactly mean “more effort”?

Equation S5. The maximum of photosynthesis rate needs to be separately defined for diatom and vegetative cells, because responses of light-harvesting systems to light intensity are completely different between diatoms and cyanobacteria. Do heterocysts show no light-intensity dependence?

Author Response

Thank you for your time and effort in reviewing the manuscript. Please see the attachment for details.

Reviewer 2 Report

The manuscript by Gao et al., outlines a modeling-based approach seeking to inform the dynamics of N and C exchange between a marine diatom – cyanobacterium association (DDA). The paper is generally well written but suffers some shortcomings. While the approach and findings are fine, the body of work as it stands is too partial a contribution for a journal of this level.

The title presents the modeling-based findings as biological reality, but the equations presented to arrive at the model use assumptions as starting point, with no linking to experimentally determined data. I could see two ways in which the outcomes could be strengthened:

1.     The equations use experimentally established data – e.g. enzyme rates, uptake and export rates published previously / determined anew; or

2.     In situ measurements are added to compare to the outcomes of modelling.

As it stands the paper is based on a circle argument – if external ammonia concentration increases, diatoms will use that and the cyanobacterial heterocysts will fix less nitrogen. As example line 93 reads: “These occur because with less NH4+, a higher fixation rate is necessary to support the N supply”, and line 99 reads “Our results suggest that DDA’s in low nutrient areas need more nutrient (C and N) transfer …..”. Put a different way, the argument in line 93 is based on human thought – what would make sense. This “common sense” is used to develop the equations, and the results of the calculations support the initial assumption. Another example is line 108 “Physiological relations like these are not considered in our model since these are complex mechanisms and are not fully understood”. Here the authors indicate that gene regulation and metabolic flux as determined by enzyme activities are what they are, NOT what our common sense would want them to be.

Specific comments:

1.     I suggest called the diazotroph here a cyanobacterium to make it clear for readers.

2.     Line 39: Replace “factor” with “question”.

3.     The sentence beginn9ng line 41 is unclear to me.

4.     Line 53: “…with the recent culture study ……” What did this study show / report?

5.     Line 228: How can you conclude this of uptake rates of ammonia are not given. How effective are ammonia uptake systems at low concentrations?

6.     Equations given in methods: It is unclear what the concepts and constants were that went into the development of these equations.

Author Response

Thank you so much for your time and effort in reviewing the manuscript. Please see the attachment.

Round 2

Reviewer 1 Report

Thank you for kind replies.

Reviewer 2 Report

Thank you for your revised manuscript, and for attending to all my concerns.